# Tension-Type Headache in Children and Adolescents

**DOI:** 10.3390/life13030825

**Published:** 2023-03-18

**Authors:** Valentina Baglioni, Silvia Orecchio, Dario Esposito, Noemi Faedda, Giulia Natalucci, Vincenzo Guidetti

**Affiliations:** Child Neurology and Psychiatry Unit, Department of Human Neuroscience, Sapienza University, Via dei Sabelli 108, 00185 Rome, Italy

**Keywords:** primary headache, tension-type headache, pediatric neurology

## Abstract

In pediatric neurology, tension-type headache (TTH) represents a very common type of primary headache during the pediatric age. Despite the high prevalence of TTH, this diagnosis is often underestimated in childhood, with relevant difficulties in the differential diagnosis of TTH from secondary and primary headache manifestations. Even among primary headaches, a clinical overlap is not so infrequent in children: migraine attacks could present tension headache-like features while tension-type headaches may display migraine-like symptoms as well. Several variables play a role in the complex trajectory of headache evolution, such as hormonal changes during adolescence, triggers and genetic and epigenetic factors. The trajectories and outcomes of juvenile migraine and TTH, as well as the transition of one form to the other, have been investigated in several long-term prospective studies. Thus, the aim of this paper is to review the current literature on the differential diagnosis workout of TTH in pediatrics, the possible outcomes during the developmental age and the appropriate therapeutic strategies. Indeed, TTH represents a challenging diagnostic entity in pediatrics, both from a clinical and a therapeutic point of view, in which early diagnosis and appropriate treatment are recommended.

## 1. Introduction

Primary headache is one of the ten most disabling medical conditions in the general population [1] and is one of the most common causes of pain among children and adolescents, with a relevant impact on the quality of life of patients and their families [2].

Headache in children and adolescents is the third most common disease-related cause of school absenteeism and can lead to impairment in the quality of life, both in family life and in leisure activities and school/work productivity [3]. Furthermore, affected young people may be at greater risk of developing both further physical problems in adulthood and psychopathological problems such as anxiety and depression [4,5,6]. In particular, it was recently estimated that children with a headache diagnosis cause considerable excess costs in public health care systems (up to 400 € per capita) [3].

Tension-type headache (TTH) is one of the most prevalent neurological disorders in the world [7,8]. Despite its relevance, this condition in pediatric neurology is often underestimated or misdiagnosed, due to its vague symptoms in childhood, with significant difficulties in the differential diagnosis with other primary or secondary headaches. Nevertheless, it is a type of primary headache that is relatively less researched in the literature, probably because of its less disabling nature than other types of primary headaches (e.g., migraine and cluster headache) [1,9,10,11].

During the pediatric age, there is often an overlap of symptoms between TTH and migraine; moreover, in the transition from childhood to adulthood, the clinical presentation of headaches often changes, usually from migraine to TTH or vice versa [6].

Thus, a timely and effective diagnostic framework and therapeutic approach is also aimed to prevent the chronicization of headaches into more severe and highly disabling forms.

In this work, we conducted a narrative review of the literature on pediatric TTH to update the characterization of this primary headache in children and adolescents. Particularly, we reported a possible diagnostic logarithm in the differential diagnosis between secondary and primary headaches with TTH during the pediatric age, because of the overlap symptoms of this headache form during development with other clinical pictures and psychopathological features. Moreover, the possible lifetime trajectories were described, reporting the results of prospective studies in the literature on children and adolescent cohorts. Finally, a current review of the therapeutic approach was conducted, both for acute and chronic treatment for the pediatric age.

## 2. Epidemiology

The estimated worldwide prevalence of TTH and migraine is around 42% in the general adult population [12]. Regarding the pediatric age, despite epidemiological data being not homogeneous, the prevalence of primary headaches has significantly increased in childhood and adolescence in recent years [13,14,15].

In pediatrics, some authors reported a prevalence of primary headaches of around 54.4% [16]. The time of onset of primary headache is rare in children younger than four, with prevalence rates increasing with age: a prevalence of 5.9–37.7% in pre-school age, 38–50% in school age and 75% in adolescence have been reported [17]. Other authors have estimated a prevalence ranging from 3–8% at 3 years up to 57–82% from 7 to 15 years [6,18].

As regards TTH specifically, epidemiological data in young subjects suggested highly variable prevalence ranging from 0.9 to 72.3%, according to different studies [15,16,19,20,21,22,23]. A recent meta-analysis and systematic review analyzed twenty-three studies reporting the prevalence of TTH in the pediatric age [14]. They reported a weighted-pooled prevalence of TTH in females of 11% and 9% in males; the prevalence of episodic TTH ranged from 4 to 29%, while chronic TTH ranged from 0.2 to 12.9% [14]. In accordance with this high variability, it should be remembered that up to 10% of headaches in preschool age remain unclassifiable or diagnosed as childhood periodic syndromes [24]. As for the gender prevalence ratio, it should be noted that TTH has a slight female preponderance of 1.2:1, much lower than the 3:1 ratio that characterizes migraine [10,14].

## 3. Differential Diagnosis

TTH is usually characterized by bilaterally localized pain, with a pressing or constricting (non-pulsating) quality, of mild to moderate intensity. The pain is not aggravated by routine physical activity (such as walking or climbing stairs) and there are usually no symptoms such as nausea, vomiting, photophobia or phonophobia. According to the International Classification of Headache Disorders 3 (ICHD-3), TTH can last from minutes to days; however, a typical episode of TTH lasts 4 to 6 h [8,21].

ICHD-3 distinguishes between different forms of TTH: infrequent episodic, frequent episodic and chronic, which substantially differ from each other in terms of frequency and duration of attacks (Table 1) [8]. However, this classification is mainly based on adult subjects.

The diagnosis of TTH is firstly aimed to exclude characteristics of secondary headache and then to differentiate it from other forms of primary headache [19,21,25,26,27].

A thorough medical history and a careful neurological objective examination cannot be ignored. The medical history should be focused on the characteristics of pain, including triggers and alleviating factors, associated symptoms (e.g., prodromes of the headache or history of bruxism), but also family and medication history and description of living environment. The neurologic physical examination should include assessment of consciousness, coordination, deep tendon reflexes, sensitivity and movement examination and visual system testing, including a fundoscopic exam. In addition, it is also important to perform a general physical examination (including an exploration of the temporomandibular joint [TMJ]) and measurement of blood pressure, weight and head circumference [21]. Neuroimaging should be considered in children with an abnormal neurologic examination or other physical findings that suggest central nervous system disease [28].

Since secondary headaches can be similar to TTH, the so-called “red flags” have been proposed, i.e., clinical or anamnestic features that suggest a secondary headache and that, if present, should prompt the clinician to carry out further investigations [27,29]. These red flags include the following:-Severe and sudden onset headache.-Very early onset of headache (i.e., in preschool age), especially considering that brain tumors are relatively more common in younger children than in older ones, while primary headaches are less common.-Focal neurological signs (e.g., ataxia or cognitive impairment), papilledema or other clinical signs suggestive of raised intracranial pressure (e.g., early morning headache, vomiting in the morning, pain disturbing sleep and headache worsened by cough or Valsalva) [26].-Atypical presentations, which should lead clinicians to suspect secondary headaches or other rarer primary headaches (e.g., cluster headaches, SUNCT, co-existing different forms of primary headaches) [29].-Headache awakening the child from sleep or consistently occurring first thing in the morning, especially if refractory to usual acute treatment. In these cases, screening for hypertension, obstructive sleep apneas, sleep bruxism or other sleep or general health conditions should also be considered [29].-Accelerated course, change in characteristics over weeks or days.-Post-traumatic headache [26].-Headache associated with personality or behavior changes.-Underlying history of neurocutaneous syndromes, systemic illness (e.g., known malignancy with possible metastases, hypercoagulopathy) or drugs or toxic substances exposure.-Headache associated with malaise and fever, which could be due to an infection. In these cases, the physical examination should include maneuvers that investigate meningeal inflammation (neck stiffness, Brudzinski and Kernig signs). However, in most cases, headache with fever is due to infections of the upper respiratory tract, such as sinusitis, based on incidental radiological findings [30,31].

Among other secondary headaches, TMJ dysfunction in children is often associated with bruxism, pain on TMJ palpation or other signs of temporomandibular dysfunction and headache with TTH-like features [32].

Among the primary headaches, the latest version of the ICHD-3 did not include any specifiers for childhood, despite the presence of distinct clinical features, compared to the adult population. Mostly the duration criteria for headache attacks could represent in pediatrics a diagnostic limit, because of the smaller duration of the single headache episode in children for migraine or TTH [25].

Moreover, in the pediatric age, patients may present an overlap or coexistence of symptoms that make it difficult to differentiate between TTH and migraine without aura. Many children with TTH report typical migraine symptoms and pain characteristics, such as one-sided pain, photophobia, phonophobia, throbbing pain and pain aggravation due to routine physical activity [19]. Conversely, migraine in childhood is more often bilateral, frontal and temporal than unilateral (35% vs. 60% in adults); the duration of the headache is often much shorter—it can last as little as 30 min—and there is a higher incidence of sensitivity to light or noise than in adults [13,33]. What makes the differentiation complex is also the difficulty of very young children to explore the clinical features of their headache that are necessary to meet the diagnostic criteria. For instance, studies have shown that 30% of children are unable to describe the quality of their pain and 16% fail to report photophobia and phonophobia [25]. Moreover, often the examiners establish the diagnosis based solely on a witness’s observation of the child’s behavior [34].

In addition, there are frequent modifications of headache characteristics over time, which may change from preschool to adolescence [35].

In fact, some authors theorized that TTH and migraine could represent a phenotypic spectrum within the primary headaches, united not only by the clinical symptomatology but also by developmental trajectories [36] and pathophysiology of chronic pain.

The pathophysiological mechanisms of TTH are based on three major components: genetic vulnerability factors, myofascial mechanisms (including myofascial nociception) and mechanisms of chronicization (including central sensitization and altered descending pain modulation) [10].

Abnormal inputs from cranial and cervical myofascial structures and, to a lesser extent, vascular structures result in the sensitization of peripheral nociceptors. This increase in nociceptive input leads to the sensitization of second-order neurons in the spinal trigeminal nucleus and dorsal horn of the spinal cord at levels C1 to C4. Aβ fibers, which normally respond only to innocuous stimuli, become pro-nociceptive or pain stimulators. Dysfunction in the modulation of descending pain pathways results in decreased inhibition and increased facilitation of nociceptive transmission. This leads to an increase in nociceptive input to the thalamus and sensitization of third-order neurons, which send impulses to the cortex. The result is a greater perception of pain and chronicity of the headache [10].

To facilitate the diagnosis and establish the most appropriate therapeutic strategy, it is useful to advise the patient to fill in a headache diary [30]. In this kind of logbook, after each headache episode, the patient should record the time of onset, duration, intensity, triggering factors, association with other symptoms, the use of any medications and their effectiveness. Figure 1 summarizes the main steps in the diagnostic approach for children with headache.

## 4. Psychopathology Features and Developmental Pathways

Primary headaches and psychopathology comorbidities may in part share pathophysiological mechanisms [37,38]. In childhood, early vulnerabilities have been underlined in the common trajectories between headache and psychopathology [39]. Recently, exposure to interpersonal violence, with abuse or neglect in early age, has been posed as a potential trigger of headache in adolescents [40] and adults [41]. In this regard, dysfunctional family relations may influence children and adolescents’ ability to cope with recurrent pain. Thus, adverse childhood experiences (ACE), with a moderate to severe stress impact during the first 18 years of life, may be negatively and cumulatively associated with chronic pain and mental health problems in youths [42], mostly expressed by anxiety and mood disorders [43]. ACEs such as psychological and/or physical abuse, loss, neglect, parents’ separation/divorce and family psychiatry history have been also reported in two out of three suicide attempts in adult life [44]. The mental health disorders associated with ACE start to be reported in adolescence until adulthood [45]. Additionally, somatoform complaints such as headache presented a higher prevalence in ACE cases. The headache prevalence has an estimated 2-fold higher increase in those cases presenting with more than two traumatic events during childhood. Additionally, the presence of chronic somatoform pain complaints (i.e., general pain; back pain) has been associated with more than one adverse childhood experience, with an increased prevalence of 1.2–1.3-fold [41,45].

Moreover, the interpersonal and parental relationship and strategies to cope with adverse situations and pain could be supportive of the negative outcome and psychopathological features in the youngest with headache. Indeed, the Interpersonal Fear Avoidance Model of Pain [46] was reported in those cases of chronic or recurrent pain and ACE in children to explain these complex mechanisms. Indeed. children’s experiences of pain may be particularly threatening, especially in the case of intense, frequent and non-predictable pain attacks. These pain experiences, recurrent within a family context where the child is somewhat unprotected and sustained, may lead to a permanent traumatic feeling of vulnerability. Fear and avoidance have been consistently associated with poor pain-related outcomes in children. Thus, recurrent primary headache may be experienced as particularly frightening for the developing child, inducing psychological distress and behavioral reactions [47]. Indeed, frequency of TTH and migraine was associated with increasing symptom scores for depression and anxiety in adolescents [48], indicating that headache-free intervals are crucial for mental health functioning. These psychopathological mechanisms of the Interpersonal Fear Avoidance Model of Pain have been reported to be sustained by a possible psychopathological mechanism in parents and the family life of a catastrophic thought [49]. As reported, the catastrophic thought associated with recurrent pain could represent a model to explain the transition through a chronic headache between childhood and adolescence [50]. Thus, those mechanisms in family life could represent a therapeutic target in psychotherapy to avoid negative outcomes and psychopathological problems [39].

A further comorbidity and therapeutic target could be represented by Alexithymia, more often reported during somatoform complaints and recurrent primary headache [51]. In adults with TTH or migraine, a higher alexithymia level was reported compared to healthy controls [43], but the association between alexithymia and primary headache in children and adolescents is not yet well investigated. In two different studies [52], a significant association between TTH and alexithymia was first reported in children presenting major alexithymia problems compared with the migraine and the control groups. Particularly, two dimensions resulted in compromised in TTH children: recognizing their own feelings and tendency for operatory thought. In accordance with these results, alexithymia seemed to create, in children, a maladaptive condition in which undistinguished feelings or emotions may undergo a process of reinforcement and become a symptom of disease. In the second study [53], significant relations were found between alexithymia and TTH, but not with migraine. They assumed that the differences between TTH and migraine may be due to differences in the primary pathogenetic mechanism of the two disorders. Genetic factors may underline the etiology of migraine more than TTH, while TTH has a more complex multifactorial pathogenesis, influenced also by the familial and social environment. In fact, primary headache is more common in families with a history of psychological disorders. Hence, empirical data suggest that adverse childhood events and psychosocial factors in interplay with genetic predispositions may lead to hyperalgesia and the development and maintenance of headaches [54]. Indeed, in addition to environmental factors, shared genetic risk factors may also have an impact on the susceptibility to both headache and psychopathological symptoms. Heritability studies on headache have mostly focused on migraines, with an estimated heritability of migraine up to 45% in twin studies [55]. Thus, genetic factors seem to play a fundamental role in this familial transmission, and new genetic risk variants have been recently identified in common between psychiatric disorders and migraines [56]. Particularly, depression and migraine may partly share underlying genetic substrates [57] even though little is yet known about the possible shared genetic risk factors for psychopathological comorbidities in headache.

In pediatric neuropsychiatry, psychopathology features have been underlined in relation to primary headache. The most recurrent comorbidities in TTH and migraine were reported with sleep disorders (TTH 8.3%; 12% migraine) and anxiety disorders (TTH 2.8%; migraine 15.6%,) [4]. In the same study, a higher rate of psychiatric disorders was reported in migraine (81.8%) in comparison with TTH (29.1%) and in the control group (8.3%), mostly represented by anxiety and depression [4].

In children with recurrent primary headache, an increased risk of developing psychiatric comorbidity were reported in adulthood [58]. In a longitudinal study, primary anxiety disorders were related to migraine but not TTH, while patients with major depression presented a 40% more likely predisposition to develop migraine [59]. Moreover, in children and adolescents, a higher prevalence of ADHD was reported in primary headache, related to the episodes’ higher frequency [60]. Indeed, in some cases, the attention deficits and hyperactivity/irritability have been reported as secondary effects due to recurrent primary headache in these children [61], with repercussions on their quality of life and school activities with further learning disabilities [62].

## 5. Developmental Trajectories in Children and Adolescents

Studying the progression over time of primary headaches is important, both from a clinical and a public health perspective. As previously mentioned, the characteristics of early-onset primary headache tend to change over time, with possible switching from one headache form to another [6].

Many studies have investigated the evolution of headache from childhood to adulthood, most of them being prospective studies on follow-up pediatric populations. Overall, approximately 25% of migraine patients were found to switch to TTH and vice versa [6]. The presence of neurodevelopmental disorders correlates with the persistence of the headache [6]. Furthermore, the male gender is more associated with remission. The consequent predominance of females during adolescence could be, in part, explained by the hormonal changes that may influence the pathophysiology of pediatric migraine, as in adult migraine. In this regard, some authors found an increased risk of headaches was found in girls who had menarche earlier [63,64,65,66], which could be explained by a common pathogenic factor such as estrogen sensitivity [67].

With aging, some triggers (such as stress) tend to be less frequent, while alcohol, smoking and neck pain usually become more prominent; a decrease in the frequency of some characteristics such as photophobia, phonophobia and vertigo was also observed [68].

In patients followed up for 10 years, it has been shown that headaches improve or regress in 60–80% of cases [69]. As for TTH specifically, it is more likely to improve in the transition from adolescence to adulthood than migraine [70].

A positive family history of headache was significantly more frequent in children with an early onset [70]. An early onset of the disease probably reflects a greater biological predisposition or increased susceptibility to specific environmental risk factors, thus potentially leading also to a worse prognosis [6,66].

Hormonal changes, different levels of stress exposure, life events and circumstances may be some of the factors that contribute to the change in the migraine phenotype over time [71,72,73]. Moreover, the prognosis of pediatric headache is adversely affected by initial diagnosis of migraine and by changes in the localization of the headache [33].

Primary headaches can indeed tend to chronicization. According to the ICHD-3 classification, chronic primary headache (CPH), tension-type or migraine-type, is defined by the presence of head pain on at least 15 days per month for more than 3 months [8].

There are little data available concerning CPH in the pediatric population and the ICHD-3 does not include notes for diagnosing pediatric CPH [8], so most of our knowledge comes from experience in adulthood.

The prevalence of CPH in children is estimated to be 0.78% to 1.5%. If the patient has been using regular and excessive headache medications for at least 3 months, the diagnosis of Medication-Overuse Headache should be considered [7]. It has been claimed that up to half of the subjects diagnosed with CPH are affected by this type of headache [74]. Most patients with this disorder improve after discontinuation of the abused drug and, moreover, respond better to prophylactic treatments.

Many studies suggest that medication abuse is lower in adolescents than in adults, this may be because adolescents are more likely to depend on their caregivers for access to medication [8,74].

## 6. Treatment Strategies

Given the high prevalence and the important impact on the quality of life of young patients, tension headache requires adequate management. Treatment of TTH is based on pain resolution (acute therapy) and the prevention of recurrence of headache episodes (chronic or prophylactic therapy).

Literature data concerning the treatment of primary headaches in childhood are still scarce, especially as regards TTH. Furthermore, there are no guidelines about the developmental age. In clinical practice, therapeutic strategies are based both on non-pharmacological therapies, including lifestyle modifications, as well as on pharmacological therapy [21].

The therapeutic choice must be weighed after a careful assessment of the patient and the impact that the headache has on the patient’s quality of life. Therefore, non-pharmacological approaches should be considered as first-line treatments.

Lifestyle recommendations should play a crucial role in headache management. Children and adolescents suffering from primary headache should be advised to maintain a balanced lifestyle including regular sleep and meals, adequate hydration, limited caffeine consumption, not smoking, very limited alcohol consumption and regular physical activities, which should be promoted to prevent metabolic disorders and to avoid stressors. In addition, the use of video games and electronic devices should be limited [75].

Among the other non-pharmacological approaches, behavioral treatments include relaxation techniques, biofeedback, cognitive behavioral therapy (CBT) or combinations of these treatments [76]. Relaxation-based therapy embraces interventions that focus on progressive muscle relaxation and similar strategies such as hypnosis. CBT comprises a multitude of interventions, such as family CBT, parent–operant strategies, multicomponent CBT and pain coping skills [76]. Finally, biofeedback techniques help patients to control functions of their autonomic nervous system (e.g., heart rate, blood pressure, muscle tension) by observing monitoring devices and reproducing desired behaviors [77].

These techniques have demonstrated efficacy in the treatment of pediatric headache [78] and are well accepted by patients and parents because of the very low risk of adverse effects in comparison with pharmacological treatments [79]. A recent interdisciplinary multimodal therapy program (which involved eight modules regarding headache education, stress management, relaxation techniques, physical activation, mindfulness and sensory training) showed reduced headache frequency and disability at six and twelve months after completion [80]. The best-known behavioral treatments for TTH are electromyographic (EMG) biofeedback and thermal biofeedback [81,82], which have shown greater responses in children than in adults [77].

Psychosocial stressors related to social and economic status, family conditions and academic performance, as well as psychiatric and somatic comorbidities and behavioral problems, can contribute to or worsen TTH. When these dynamics are detected, a biopsychosocial approach to care should be adopted [45,83]. Such an approach may also be useful considering that children with TTH in some cases live with negative parental relationships or in disadvantaged family environments from emotional, economic, or socio-cultural points of view [83,84]. Management of stressors, anxiety and mood disorders should take precedence over pharmacological management of headache [27].

Acute pharmacological therapy is based on paracetamol and NSAIDs. In particular, paracetamol should be the first-line medicine, feasible even for the youngest [85]. On the other hand, despite being recommended for the treatment of pediatric migraine, ibuprofen has not obtained the same consensus in TTH [27].

Other NSAIDs (ketoprofen, diclofenac, naproxen) have shown efficacy in the treatment of headache in adults, but definitive data on their use in pediatric TTH are not available [86].

Prophylactic therapy aims to reduce the frequency, intensity and duration of headache episodes, with an improvement in the quality of life. Pharmacological approaches should be considered only if non-pharmacological therapies have not been effective [27,85,87].

There are currently no clear recommendations for the prophylactic treatment of pediatric patients with primary headache [88]; the use of preventive drugs in pediatric tension headaches is therefore completely “off-label” [26,87].

Amitriptyline, in relatively low doses, is generally the first choice for TTH prophylaxis [89]. Common side effects include drowsiness, weight gain, dry mouth, dizziness, sweating, constipation and increased appetite [10].

In a prospective double-blind study of patients aged 14 to 76 years, valproate was superior to placebo in the prophylactic treatment of chronic primary headache (albeit more so in chronic migraine than in chronic tension-type headache) [90].

Tolerability issues of valproate include dizziness, abdominal pain, nausea, somnolence, tremor, impotence and hair loss [90].

More recently, the efficacy of magnesium as a prophylactic treatment for TTH has been considered, with a significant improvement in symptoms and reduction in the frequency of attacks, thus improving the disabling aspects of the headache itself [87,91]. Table 2 summarizes the treatment approaches for TTH in children and adolescents.

## 7. Conclusions

Tension-type headache represents a very common disorder in children and adolescents. In our review, we underlined the careful differential diagnosis required between TTH and both secondary headaches and other primary headaches. Red flags in pediatrics could be suggestive of secondary features of headache, and should be carefully considered for a rapid and effective therapeutic approach, such as for a very early onset in pre-school age or behavioral changes and irritability. Moreover, TTH can result in a real challenge diagnosis due to the overlap of symptoms with other primary headaches, particularly with migraine without aura, because of similar episodes lengths, pain localization and associated symptoms. Mostly, it is important to highlight the possible developmental trajectory from TTH to migraine and vice versa in 20–25% of cases, as reported by prospective cohort studies. Putative neurophysiological mechanisms in pain chronicization and psychopathological risk factors have been widely considered as common mechanisms in these primary headaches.

The correct management includes a thorough history and physical examination, without leaving out delving into possible psychological disorders in patients, which can play an important role in headache, as for mood or anxiety disorders. An accurate diagnosis allows us to implement the appropriate therapeutic strategies, which consist in pediatrics of lifestyle modification and non-pharmacological and pharmacological treatments. In this way, we have the possibility to improve the quality of life of our young patients and their families.

## Figures and Tables

**Figure 1 life-13-00825-f001:**
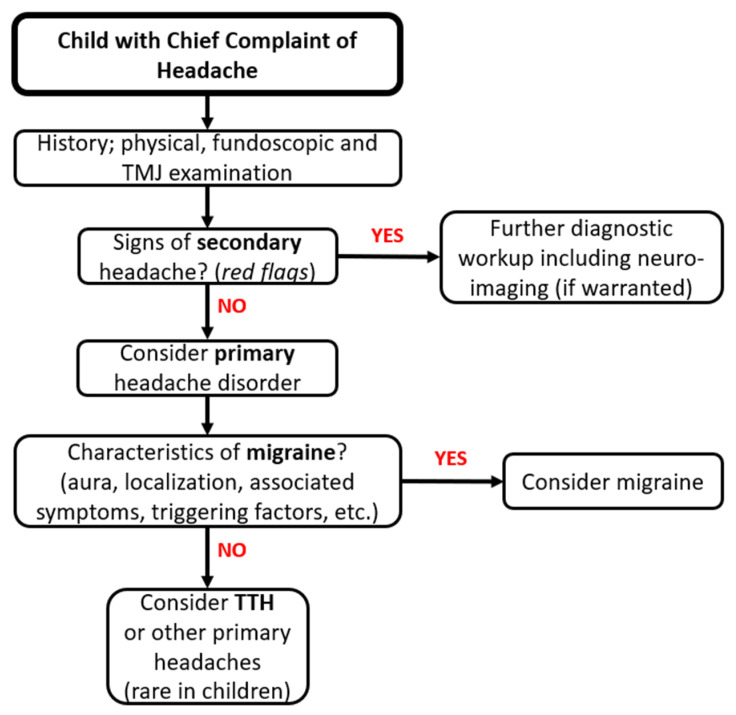
Schematic approach to diagnosis of headache in children. TMJ, temporomandibular joint; TTH, tension-type headache.

**Table 1 life-13-00825-t001:** Diagnostic criteria of tension-type headache subtypes (adapted from [8]).

Infrequent Episodic TTH	Frequent Episodic TTH	Chronic TTH
**A.** At least 10 episodes of headache occurring on <1 day per month on average (<12 days per year) and fulfilling criteria B–D	**A.** At least 10 episodes of headache occurring on 1–14 days per month on average for >3 months (12 and <180 days per year) and fulfilling criteria B–D	**A.** Headache occurring on 15 days per month on average for >3 months (180 days per year), fulfilling criteria B–D
**B.** Lasting from 30 min to 7 days		**B.** Lasting hours to days, or unremitting
**C.** At least two of the following four characteristics: 1. Bilateral location; 2. Pressing or tightening (non-pulsating) quality; 3. Mild or moderate intensity; 4. Not aggravated by routine physical activity such as walking or climbing stairs.		
**D.** Both of the following: 1. No nausea or vomiting; 2. No more than one of photophobia or phonophobia.		**D.** Both of the following: 1. No more than one of photophobia, phonophobia or mild nausea; 2. Neither moderate or severe nausea nor vomiting.
**E.** Not better accounted for by another ICHD-3 diagnosis.		

**Table 2 life-13-00825-t002:** Tiered treatment approaches to tension-type headache in children and adolescents. CBT, cognitive behavioral therapy; h, hour; y, year.

Treatment Approaches		Ref.
First line: Lifestyle modifications	Hydration (recommended 4–8 y, 1.2 L; 9–13 y, 1.6–1.8 L; 14–18 y, 1.8–2.6 L) Sleep (recommended at least 8 h/night) Diet (consistent, well-balanced meals, limitation of caffeine) Stress avoidance Electronic devices (avoid or limit screen time) Regular Exercise Avoidance of smoking, alcohol drinking, other abuse substances	[75]
Second line: Non-pharmacological treatments	Behavioral therapy (CBT, biofeedback, stress management techniques) Other complementary treatments	[76,79]
Third line: Pharmacological therapy	Acute/abortive treatment: Paracetamol (15–20 mg/kg every 4–6 h; max. 90 mg/kg/day) Ibuprofen (if <12 y, 10 mg/kg every 6–8 h; if >12 y, 400–600 mg every 6–8 h; max. 1800 mg/day)	[27]
	Preventive/prophylaxis: Amitriptyline (1 mg/kg/day) Valproate (10–20 mg/kg/day) Magnesium (200–300 mg, twice per day)	[89] [90] [91]

## Data Availability

Data sharing is not applicable to this article as no new data were created or analyzed in this study.

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
