# Peer review of "Tension-Type Headache in Children and Adolescents"

_life, 2023, doi:10.3390/life13030825_

Round 1

Reviewer 1 Report

The topic of the review is important and covers an underdiagnosed and often improperly treated area. The authors list a number of therapeutic approaches that reflect the poverty of the extension of therapies used in adults to the adolescent phase.

The reference section is in need of updating as it is somewhat outdated. I suggest the following references that have appeared in recent years regarding the topic.

Epidemiology: PMID: 36782182, PMID: 36085007, PMID: 34074243, PMID: 32873227

Burden: PMID: 33849431

Neuroimaging: PMID: 34229614, PMID: 34294048

Costs: PMID: 34800970

Therapy: PMID: 36008766

Author Response

We thank the reviewer for the suggestions and for updated references. We added most of the proposed paper in our reference section and commented in the text appropriately. Neuroimaging papers were beyond the scope of this paper and were not included.

Reviewer 2 Report

In this manuscript, the authors review the current literature on the differential diagnosis of childhood tension-type headache, possible outcomes during development, and appropriate therapeutic strategies. They also draw attention to the importance of early diagnosis and appropriate treatment.

The topic is timely and may attract much attention.

However, I have some suggestions to improve this paper:

1. The types of tension-type headache (episodic, chronic) should be presented in more detail.

2. The pathomechanism of tension-type headache should be better explained (current knowledge).

3. A description of behavioral treatments, relaxation techniques, biofeedback, and cognitive-behavioral therapy would be useful.

4. Comparing the therapeutic options for children, adolescents, and adults would be interesting. Is there a difference? Why can't the same solutions be applied?

5. In general, I recommend authors use more references to back their claims. I believe that adding more citations will help to provide better and more accurate background to this study.

Author Response

1. Thank you for pointing this out. We agree with this comment. Therefore, we have included a new Table 1 to describe in detail the diagnostic differences among episodic and chronic TTH.

2.Thank you for this suggestion. It would have been interesting to explore this aspect. However, in the case of our study, it seems slightly out of scope because we decided not to focus too much on pathophysiology of TTH but mainly on the clinical aspects in pediatrics. However, we have included more details about the general mechanisms in the third section of our paper.

  3. Agree. We have, accordingly, modified section 6 to emphasize this point   4.You have raised an important point here. However, literature is lacking in giving detailed and definite explanation on the reasons of different clinical response between children, adolescents and adults and we believe that speculating on this topic in the limited space we have left would not be effective.

5. We agree and have incorporated more references, which have also been suggested by other reviewers.

Reviewer 3 Report

Thank you for the report, good work. However it would be interesting to include some notes on Headache attibuted to temporomandibular disorder (ICHD-III 11.7). For example, including bruxism in medical history and a TMA exploration in every headache consultation. This headache is mentioned in line 121 but it is not described how to identify it. 

Moreover, including fundoscopic and TMA examination in figure 1 (history, physical examination) is recommended. 

Author Response

We appreciated the suggestions and modified both text and figure 1 to include more details on headache attributed to temporomandibular disorder.

Reviewer 4 Report

A well written paper

a few suggestions 

1. In the discussion:

discuss migraines , SUNCT and cluster headaches should ne considered if having atypical features 

also consider adding these headaches can have co existing forms of headaches 

On the table : CT is highly discouraged as radiation causes long term problems in pediatrics 

For wake up headaches and refractory talk about screening for HTN sleep apnea etc

Agree with lifestyle: consider adding exercise 

for amytriptyline add side effects and interactions 

good paper

may add ise low dose  peds compatible scanner if mri cannot be done but mri is preferred 

Author Response

We thank the reviewer for the relevant comments. We modified the paper accordingly.

In addition to the above comments, some spelling and punctuation errors have been corrected.